# A Two-Stage Data Association Approach for 3D Multi-Object Tracking

**DOI:** 10.3390/s21092894

**Published:** 2021-04-21

**Authors:** Minh-Quan Dao, Vincent Frémont

**Affiliations:** LS2N, CNRS, École Centrale de Nantes, 1 Rue de la Noë, 44321 Nantes, France; minh-quan.dao@ec-nantes.fr

**Keywords:** multi-object tracking, data association, autonomous vehicles

## Abstract

Multi-Object Tracking (MOT) is an integral part of any autonomous driving pipelines because it produces trajectories of other moving objects in the scene and predicts their future motion. Thanks to the recent advances in 3D object detection enabled by deep learning, track-by-detection has become the dominant paradigm in 3D MOT. In this paradigm, a MOT system is essentially made of an object detector and a data association algorithm which establishes track-to-detection correspondence. While 3D object detection has been actively researched, association algorithms for 3D MOT has settled at bipartite matching formulated as a Linear Assignment Problem (LAP) and solved by the Hungarian algorithm. In this paper, we adapt a two-stage data association method which was successfully applied to image-based tracking to the 3D setting, thus providing an alternative for data association for 3D MOT. Our method outperforms the baseline using one-stage bipartite matching for data association by achieving 0.587 Average Multi-Object Tracking Accuracy (AMOTA) in NuScenes validation set and 0.365 AMOTA (at level 2) in Waymo test set.

## 1. Introduction

Multi-object tracking have been a long standing problem in computer vision and robotics community since it is a crucial part of any autonomous systems. From the early work of tracking with hand-craft features, the revolution of deep learning which results in highly accurate object detection models [1,2,3] has shifted the focus of the field to the track-by-detection paradigm [4,5]. In the framework of this paradigm, tracking algorithms receive a set of object detection, usually in the form of bounding boxes, at each time step and they aim to link detection of the same object across time to form trajectories.

While image-based methods of this paradigm have reached a certain maturity, 3D tracking is still in its early phase where most of the published approaches are originated from successful 2D exemplars. One popular method is [6] which extends [4] into 3D space. In these works, detections are linked to tracks by solving a bipartite matching with the Hungarian algorithm [7], then states of tracks are updated by a Kalman filter. Taking a similar approach to establishing detection-to-track correspondence, [8] trains a network for calculating the matching cost instead of using the 3D Intersection over Union (IoU). In [9,10], objects’ poses in the current and several future frames are predicted by deep neural networks. Thus, tracks can be formed by greedy closest-point matching.

Even though 3D tracking has been progressed rapidly thanks to the availability of standardized large scale benchmarks such as KITTI [11], NuScenes [12], Waymo Open Dataset [13], the focus of the field is placed on developing better object detection models rather than developing better tracking algorithm as shown in Table 1 which presents the performance measured by the AMOTA metric of tracking algorithms following the track-by-detection paradigm and the performance of their object detector measured by mean Average Precision (mAP). AMOTA is a scalar value representing how well the algorithm does in limiting:ID switches (IDS): the number of times tracks are associated with wrong detections;False Positives (FP): the number of times real objects are missed detected;False Negatives (FN): the number of times the tracking algorithm reports tracks in places where there are no real objects present.

There are two trends that can be observed in this table. First, tracking performance experiences a boost when a better object detection model is introduced. Second, the method of AB3DMOT [6], which uses the Hungarian algorithm on some metrics (e.g., 3D IoU, Mahalanobis distance) to perform data association, Kalman Filters to update tracks’ states once they have associated detections, and set of heuristic rules to manage birth and death of tracks, is favored by most recent 3D tracking systems.

The reason for AB3DMOT’s popularity is that despite its simplicity, it achieves competitive result in challenging datasets at significantly high frame rate (more than 200 FPS). However, such simplicity comes at the cost of the MOT system being vulnerable to false associations due to occlusion or imperfect detections which is case for objects in a clutter or far away from the ego vehicle.

Aware of the shortage of a generic 3D tracking algorithm which can better handle the occlusion and imperfect detections, yet remains relatively simple, we adapt the image-based tracking method proposed by [22] to the 3D setting. Specifically, this method is a two-stage data association scheme. In this scheme, each tracked trajectory is called a tracklet and is assigned a confidence score computed based on how well associated detection matches with tracklet. The first association stage aims to establish the correspondence between high-confident tracklets and detection. The second stage matches the left over detection with the low confident tracklets as well as link low-confident tracklets to high-confident ones if they meet a certain criterion.

In this paper, we make two contributions
Our main contribution is the adaptation of an image-based tracking method to the 3D setting. In details, we exploit a kinematically feasible motion model, which is unavailable in 2D, to facilitate the prediction of objects’ poses. This motion model defines the minimal state vector needed to be tracked.Extensive experiment carried out in various datasets proves the effectiveness of our approach. In fact, our better performance, compared to AB3DMOT-style models, show that adding a certain degree of re-identification can improve the tracking performance while keeping the added complexity to the minimum.Our implementation is available at https://github.com/quan-dao/track_with_confidence accessed on 21 April 2021.

## 2. Related Work

A multi-object tracking system in the track-by-detection paradigm consists of an object detection model, a data association algorithm and a filtering method. While the last two components are domain agnostic, object detection models, especially learning-based methods, are tailored to their operation domain (e.g images or point clouds). This paper targets autonomous driving where objects’ poses are required thus being interested in 3D object detection models. However, developing such a model is not in the scope of this paper, instead we use the detection result provided by baseline models of benchmarks (e.g., PointPillars of NuScenes) to focus on the data association algorithm and to have a fair comparison. Interested readers are referred to [23] for a review of 3D object detection.

Data association via the Hungarian algorithm was early explored in [24] where a 2-stage tracking scheme was proposed for offline 2D tracking. Firstly, detections are linked frame-by-frame to form tracklets by associate detections to tracklets via solving a LAP with the Hungarian algorithm. The cost matrix of this LAP is computed based on geometric and appearance cue. While the geometric cue is the 2D IoU, the appearance cue is the correlation between two bounding boxes. Secondly, tracklets are associated with each other to compensate trajectory fragments and ID switches due to occlusion. Similar to the previous step, this association is also formulated as a LAP and solved by the Hungarian algorithm.

Due to its batch-processing nature, the method of [24] cannot be applied to online tracking. The authors of [4] overcomes this by eliminating the second stage, which effectively sacrifices the ability of re-identifies objects after a period of occlusion. Despite its simplicity, SORT — the method proposed by [4] – achieves competitive result in MOT15 [25] with lightning-fast inference speed (260 Hz). The success of SORT inspired [6] to adapt it to 3D setting by using 3D IoU as the affinity function. The performance of SORT in 3D setting is later improved in [15] showing the superiority over 3D IoU of the Mahalanobis distance which is the magnitude of difference between the expected detection given the ego vehicle pose and the real detection while taking into account their uncertainty. In [26], the authors integrate the 3D version of SORT into a complete perception pipeline for autonomous vehicles.

The two-stage association scheme is adapted to online tracking in [22] which proposes a confidence score to quantify tracklets quality. Based on this score, tracklets are associated with detections or another tracklets, or terminated. The appearance model learned by ILDA in [22] is improved by deep learning in the follow-up work [27]. Recently, this association scheme is revisited in the context of image-based pedestrian tracking by [28] which proposed to use the rank of the Hankel matrix as tracklets motion affinity. To be specific, this technique estimates a tracklet ’s dynamic by a linear regressor taking its previous states as input. In noise-free scenarios, the order of such a regressor (i.e., the number of past states needed to estimate the current state) is equal to the rank of the Hankel matrix which formula can be found in [28]. The intuition behind this technique is that if two tracklets belong the same trajectory, explaining their merged trajectory would require a low order regressor. This technique is popular in image-based tracking despite being prone to deterioration due to noise because of the absence of an accurate motion model in this space. However, objects’ motion in 3D can be well explained by their kinematic models. Therefore, our approach employs two different kinematic models for two different categories of objects to have more computationally efficient and accurate motion affinity.

Differently from [22] and its related works, this paper applies the two-stage association scheme to online 3D tracking. In addition, we can provide competitive result despite relying solely on geometric cue to compute tracklet affinity by exploiting the Constant Turning Rate and Velocity (CTRV) motion model which can accurately predict objects position in 3D space by exploiting their kinematic.

## 3. Method

### 3.1. Problem Formulation

Online MOT in the sense of track-by-detection aims to gradually grow the set of tracklets $T_{0:t}={\left\{T^{i}\right\}}_{i=1}^{|T_{0:t}|}$ by establishing correspondences with the set of detections received at every time step $D_{t}={\left\{d_{t}^{j}\right\}}_{j=1}^{\left|D\right|}$ and updating tracklets state accordingly. A detection $d_{t}^{j}$ at time step *t* encapsulates information of an object as a 3D bounding box
(1)$$d_{t}^{j}={\left[\begin{array}{ccccccc} x & y & z & \theta & w & l & h \end{array}\right]}^{T},$$
here, [x,y,z] is the position of the box’s center, θ is its heading direction, and [w,l,h] is its size. It is worth noticing that in the context of autonomous driving, objects are assumed to remain in contact with the ground; therefore, their detections are up-right bounding boxes which orientation is described by a single number — the heading angle. A tracklet is a collection of state vectors corresponding to the same object $T^{i}=\left\{x_{k}^{i}|0\leq t_{s}^{i}\leq k\leq t_{e}^{i}\leq t\right\}$, here $t_{s}^{i},t_{e}^{i}$ are respectively the starting- and ending-time of the tracklet.

The correspondence between $T_{0:t}$ and $D_{t}$ can be formally defined as finding the set $T_{0:t}^{*}$ that maximizes its likelihood given $D_{t}$.
(2)$$T_{0:t}^{*}=\underset{T_{0:t}}{\mathrm{argmax}}\;p\left(T_{0:t}|D_{t}\right)$$

Due the exponential growth of possible associations between $T_{0:t}$ and $D_{t}$, Equation (Equation 2) is computationally intractable after a few time steps. In this paper, such a correspondence is approximated by the two-stage data association proposed by [22] as shown in the following.

### 3.2. Two-Stage Data Association

#### 3.2.1. Tracklet Confidence Score

The reliability of a tracklet is quantified by a confidence score which is calculated based on how well associated detections match with its states across its life span and how long its corresponding object was undetected.
(3)$$\mathrm{conf}\left(T^{i}\right)=\left(\frac{1}{L^{i}}\sum_{k\in \left[t_{s}^{i},t_{e}^{i}\right]|v^{i}\left(k\right)=1}\Lambda \left(T^{i},d_{k}^{j}\right)\right)\times \exp\left(-\beta \frac{W}{L_{i}}\right)$$
where $v^{i}\left(k\right)$ is a binary indicator which takes 1 if the tracklet has a detection $d_{k}^{j}$ associated with it at time step *k*, and 0 otherwise. $L^{i}$ is the number of time step that the traklet gets associated with a detection. Λ(·) is the affinity function which evaluates the similarity between a track and a detection. Its detail will be presented in the following subsection. β is a tuning parameter which takes high value if the object detection model is accurate. $W=t-t_{s}^{i}-L^{i}+1$ is the number of time step that tracklet was undetected (i.e., did not have associated detection) calculated from its birth to the current time step *t*.

Applying a threshold $\tau^{c}$ this confidence score divides the set $T_{0:t}$ into a subset of high-confident tracklets $T_{0:t}^{h}=\left\{T^{i,h}|\mathrm{conf}\left(T^{i}\right)>\tau^{c}\right\}$ and a subset of low-confident tracklets $T_{0:t}^{l}=\left\{T^{i,l}|\mathrm{conf}\left(T^{i}\right)\leq \tau^{c}\right\}$. These two subsets are the fundamental elements of the two-stage association pipeline showed in Figure 1

#### 3.2.2. Affinity Function

Affinity function Λ(·) is to compute how similar a detection to a tracklet or a tracklet to another. As mentioned earlier, due to the lack of colorful texture in point cloud, the affinity function used in this work is just comprised of geometric cue. Specifically, it is the sum of position affinity and size affinity.
(4)$$\Lambda \left(T^{i},d_{t}^{j}\right)=\Lambda {\left(T^{i},d_{t}^{j}\right)}^{p}+\Lambda {\left(T^{i},d_{t}^{j}\right)}^{s}$$

The scheme for computing position affinity between a tracklet and a detection or between two tracklets are shown in Figure 2

As shown in Figure 2a, the position affinity $\Lambda {\left(T^{i},d_{t}^{j}\right)}^{p}$ between a tracklet $T^{i}$ and a detection $d_{t}^{j}$ is defined as the Mahalanobis distance between tracklet’s last state propagated to the current time step *t* and the measurement vector $z_{t}^{j}$ extracted from $d_{t}^{j}$
(5)$$\Lambda {\left(T^{i},d_{t}^{j}\right)}^{p}={\left(h({\bar{x}}_{e}^{i})-z_{t}^{j}\right)}^{T}\cdot S^{-1}\cdot \left(h({\bar{x}}_{e}^{i})-z_{t}^{j}\right)$$
where ${\bar{x}}_{e}^{i}$ is last state of tracklet $T^{i}$ propagated to the current time step using the motion model which will be presented below. h(·) is the measurement model computing the expected measurement using the inputted state and the measurement vector $z_{t}^{j}$ extracted from $d_{t}^{j}$
(6)$$z_{t}^{j}={\left[x,y,z,\theta \right]}^{T}$$

The matrix S is the covariance matrix of the innovation (i.e., the difference between expected measurement $h\left({\bar{x}}_{e}^{i}\right)$ and its true value $z_{t}^{j}$)
(7)$$S=H\cdot P\cdot H^{T}+R$$
here, H=δh/δx is the Jacobian of the measurement model. P,R are covariance matrix of ${\bar{x}}_{e}^{i}$ and $z_{t}^{j}$, respectively. These covariance matrices are calculated based on training data using the approach proposed by [15].

In the case of two tracklets $T^{i}$ and $T^{j}$, assuming $T^{j}$ starts after $T^{i}$ ended, their motion affinity is, according to Figure 2b, is the sum of
Mahalanobis distance between the last state of $T^{i}$ propagated forward in time and the first state of $T^{j}$;Mahalanobis distance between the first state of $T^{j}$ propagated backward in time and the last state of $T^{i}$.
(8)$$\Lambda {\left(T^{i},T^{j}\right)}^{p}=\Lambda {\left(T^{j},{\bar{x}}_{e}^{i}\right)}^{p}+\Lambda {\left(T^{i},{\bar{x}}_{s}^{j}\right)}^{p}$$
here, ${\bar{x}}_{e}^{i}$ is the last state of tracklet $T^{i}$ propagated forward in time to the first time step of tracklet $T^{j}$, while ${\bar{x}}_{s}^{j}$ is the first state of tracklet $T^{j}$ propagated backward in time to the last time step of tracklet $T^{i}$. The size affinity $\Lambda {\left(T^{i},d_{t}^{j}\right)}^{s}$ is computed as follows:(9)$$\Lambda {\left(T^{i},d_{t}^{j}\right)}^{s}=-\frac{|w_{e}^{i}-w_{t}^{j}|}{w_{e}^{i}+w_{t}^{j}}\cdot \frac{|l_{e}^{i}-l_{t}^{j}|}{l_{e}^{i}+l_{t}^{j}}\cdot \frac{|h_{e}^{i}-h_{t}^{j}|}{h_{e}^{i}+h_{t}^{j}}$$
here, $\left[w_{e}^{i},l_{e}^{i},h_{e}^{i}\right]$ are the size of the last state of tracklet $T^{i}$, while $\left[w_{t}^{j},l_{t}^{j},h_{t}^{j}\right]$ are the size of the detection $d_{t}^{j}$. In the case of two tracklets $T^{i}$ and $T^{j}$, assuming $T^{j}$ starts after $T^{i}$ ended, their size affinity is
(10)$$\Lambda {\left(T^{i},T^{j}\right)}^{s}=-\frac{|w_{e}^{i}-w_{s}^{j}|}{w_{e}^{i}+w_{s}^{j}}\cdot \frac{|l_{e}^{i}-l_{s}^{j}|}{l_{e}^{i}+l_{s}^{j}}\cdot \frac{|h_{e}^{i}-h_{s}^{j}|}{h_{e}^{i}+h_{s}^{j}}$$

The subscript e,s in Equation (Equation 10) respectively denote the ending and starting state of a tracklet.

To reduce the risk of false association, a threshold is applied to the affinity
(11)$$\Lambda \left(T^{i},d_{t}^{j}\right)=\left\{\begin{array}{c} \Lambda \left(T^{i},d_{t}^{j}\right),\;\;\mathrm{if}\;\Lambda \left(T^{i},d_{t}^{j}\right)<\sigma \\ \infty ,\;\;\mathrm{otherwise}\; \end{array}\right.$$

#### 3.2.3. Local Association

In this association stage, high-confident tracklets ($T_{0:t}^{h}$) are extended by their correspondence in the set of detections $D_{t}$. This tracklet-to-detection is found by solving the linear assignment problem characterized by the cost matrix L as follows:(12)$$L=\left[\begin{array}{c} l_{i,j} \end{array}\right]\in R^{h\times d},\;\mathrm{with}\;l_{i,j}=-\Lambda \left(T^{i,h},d_{t}^{j}\right),T^{i,h}\in T_{0:t}^{h}$$
where h,d are respectively the number of high-confident tracklets and the number of detections. The intuition of this association stage is that because tracklets with high-confident have been tracked accurately for several time steps, the affinity function can identify if a detection is belong to the same object as the tracklet with high accuracy, thus limiting the possibility of false correspondences. In addition, low-confident tracklets are usually resulted from fragment trajectories or noisy detections, excluding them from this association stage help reduces the ambiguity.

#### 3.2.4. Global Association

As shown in Figure 1, the global association stage carries out the following tasks

Matching low-confident tracklets with high-confident ones;Matching low-confident tracklets with detections left over by the local association stage;Deciding to terminate low-confident tracklets.

These tasks are simultaneously solved as a LAP formulated by the following cost matrix
(13)$$G_{\left(l+d^{\prime }\right)\times \left(h+l\right)}=\left[\begin{array}{cc} A_{l\times h} & B_{l\times l}\\ \infty_{d^{\prime }\times h} & C_{d^{\prime }\times l} \end{array}\right]$$
here, l,d are respectively the number low-confident tracklets and detections left over by the local association stage. $\infty_{d^{\prime }\times h}$ is the matrix of size $d^{\prime }\times h$ with every element is set to *∞*. Recall *h* is the number of high-confident tracklets. Submatrix A is the cost matrix of the event where low-confident tracklets are matched with high-confident ones
(14)$$A=\left[a_{i,j}\right]\in R^{l\times h},\;\mathrm{with}\;a_{i,j}=-\Lambda (T^{i,l},T^{j,h})$$

Submatrix B represents the event where low-confident tracklets are terminated.
(15)$$B=\left[b_{i,j}\right]\in R^{l\times l},\;\mathrm{with}\;b_{i,j}=\left\{\begin{array}{c} -\log\left(1-\mathrm{conf}\left(T^{i}\right)\right),\;\mathrm{if}\;i=j\\ \infty ,\;\mathrm{otherwise} \end{array}\right.$$

Finally, submatrix C is the cost of the associating low-confident tracklets with detections left over by local association stage.
(16)$$C=\left[c_{i,j}\right]\in R^{d^{\prime }\times l},\;\mathrm{with}\;c_{i,j}=-\Lambda (T^{j},d_{t}^{i})$$

The solution to the LAP in this stage and in the Local Association stage is the association that *minimize* the cost and can be either found by the Hungarian algorithm for the optimal solution or by a greedy algorithm which iteratively picks and removes correspondence pair with the smallest cost until there is no pair has cost less than a threshold. The detail of this greedy algorithm can be found in [15] or in Section 3.4.

Once a detection is associated with a tracklet, its position and heading is used to update the tracklet’s state according to the equation of the Kalman Filter, while its sizes is averaged with tracklet’s sizes in the past few time steps to result in updated sizes. Detections do not get associated in the global association stage are used to initialize new tracklets.

### 3.3. Motion Model and State Vector

Exploiting the fact that objects are tracked in 3D space of a common static reference frame which can be referred to as the world frame, motion of objects can be described by more kinematically accurate models, compared to the commonly used Constant Velocity (CV) model. In this work, we use the Constant Turning Rate and Velocity (CTRV) model to predict motion of car-like vehicles (e.g., cars, buses, trucks), while keeping the CV model for pedestrians.

For car-like vehicles, its state can be described by
(17)$$x={\left[x,y,z,\theta ,v,\dot{\theta },\dot{z}\right]}^{T}$$
here, [x,y,z] is the location in the world frame of the center of the bounding box represented by the state vector, θ is the heading angle, v is longitudal velocity (i.e., velocity along the heading direction), $\dot{\theta },\dot{z}$ are respectively velocity of θ and *z*.

The motion on x-y plane of car-like vehicles can be predicted using CTRV as follows:(18)$$x_{t+1}=x_{t}+\left[\begin{array}{c} \frac{v}{\dot{\theta }}\left(\sin\left(\theta +\dot{\theta }\Delta t\right)-\sin\left(\theta \right)\right)\\ \frac{v}{\dot{\theta }}\left(-\cos\left(\theta +\dot{\theta }\Delta t\right)+\cos\left(\theta \right)\right)\\ \dot{z}\Delta t\\ \dot{\theta }\Delta t\\ 0\\ 0\\ 0 \end{array}\right]$$
where Δt is the sampling time. Please note that in Equation (Equation 18), *z* is assumed to evolve with constant velocity. In the case of zero turning rate (i.e., $\dot{\theta }=0$),
(19)$$x_{t+1}=x_{t}+{\left[\begin{array}{ccccccc} v\cos(\theta ) & v\sin(\theta ) & \dot{z}\Delta t & \dot{\theta }\Delta t & 0 & 0 & 0 \end{array}\right]}^{T}$$

The state vector of a pedestrian is
(20)$$x={\left[\begin{array}{cccccccc} x & y & z & \theta & \dot{x} & \dot{y} & \dot{z} & \dot{\theta } \end{array}\right]}^{T}$$

The motion of pedestrians is predicted according to CV model
(21)$$x_{t+1}=x_{t}+{\left[\begin{array}{cccccccc} \dot{x} & \dot{y} & \dot{z} & \dot{\theta } & 0 & 0 & 0 & 0 \end{array}\right]}^{T}\cdot \Delta t$$

### 3.4. Complexity Analysis

As shown in Figure 1, our data association pipeline is made of four components: Local Association, Global Association, Update Tracklets’ States, Update Tracklets’ Confidence. This section gives an analysis of the time complexity referred to as complexity of these four components.

Let *d* and *h* be the number of detections and the number of high confident tracklets, respectively. The time complexity of the Local Association step is the sum of the complexity of computing the cost matrix L in Equation (Equation 12) and solving the LAP represented by L. Since L has the size of h×d, the complexity of computing L is O(hd).

The LAP represented by L can be solved by either the Hungarian algorithm or a greedy algorithm [15]. The complexity of the Hungarian algorithm is $O(hd^{2})$. On the other hand, the greedy algorithm is made of two steps presented in Algorithm 1.
**Algorithm 1:** Greedy algorithm for solving LAP
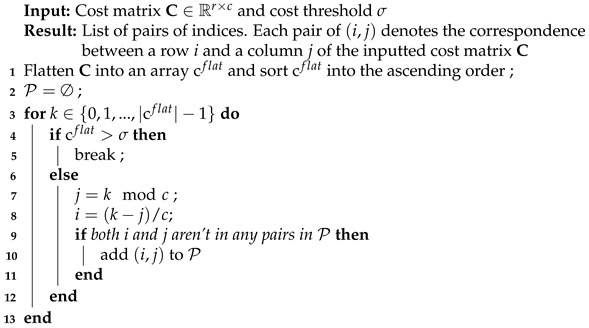


The first step of sorting the flattened cost matrix $C\in R^{r\times c}$ has the complexity of O(rclog(rc))=O(rclog(c)) assuming c>r. The complexity of the second step in the best case scenario where the for loop is stopped at k=0, meaning there is no valid association, is O(1). The worst case scenario happens when the For Loop proceeds till the last value of *k*, which means every possible association has its affinity less than the threshold σ. In this case, the complexity is $O(|c^{flat}|)=O\left(rc\right)$. As the result, the complexity of the greedy algorithm is
(22)O(rclog(c))+O(rc)=O(rclog(c))

Using Equation (Equation 22), the complexity of the Local and Global Association step solved by the greedy algorithm are O(hdlog(d)) and $O\left(\left(l+d^{\prime }\right)\left(h+l\right)\right)\log\left(l+d^{\prime }\right)$, respectively. Recall *l* and $d^{\prime }$ are the number of low-confident tracklets and the number of detections left over by the Local Association step.

The other steps, Update Tracklets’ States and Update Tracklets’ Confidence, have the linear complexity because they are made of one loop through all tracklets.

## 4. Experiments

The effectiveness of our method is demonstrated by benchmarking against SORT-style baseline models on three large scale datasets: KITTI, NuScenes, and Waymo. In addition, we perform an ablation study using NuScenes dataset to better understand the impact of each component on our system’s general performance.

### 4.1. Tuning the Hyper Parameters

There are three hyper parameters in our data association pipeline: the confidence threshold $\tau^{c}$, the detection model accuracy β in Equation (Equation 3), and the affinity threshold σ.

The confidence threshold $\tau^{c}$ is set to 0.5 according to [22]. It is worth noticing that [22] suggests that this parameter does not have any significant effect on the tracking performance. The value of β is chosen empirically such that a high-confident tracklet becomes low-confident after being undetected for three frames.

As observed from experiments, the position affinity $\Lambda {\left(\cdot ,\cdot \right)}^{p}$ is the dominant component in the tracklet-to-detection and tracklet-to-tracklet affinity. Since the position affinity, which is the Mahalanobis distance between expected detection and real detection, is $\chi^{2}$ distributed, the affinity threshold σ in Equation (Equation 11) is chosen according to the percentile of $\chi^{2}$ distribution where the position affinity resulted from a correct association is expected to fall into. Notice that the degree of freedom of the $\chi^{2}$ distribution of our interest is 4 due to the dimension of the measurement vector z in Equation (Equation 6).

Intuitively, the affinity threshold σ determines how conservative our tracking algorithm is. Small σ makes our algorithm be more skeptical by rejecting detections that are close, but not close enough to the prediction of tracks’ states. This works well in the scenario where a large number of false-positive detections presents (e.g., Waymo dataset). However, too small σ can reject correct detections thus deteriorating the tracking performance. The method used for searching for a good value of σ is

Performs a coarse grid search with the expected percentile of $\chi^{2}$ distribution in the set {10%,50%,90%,95%,97.5%,99%} which means the value of σ is in the set {0.53,1.67,3.89,4.75,5.57,6.64}, while keeping the rest of hyper parameters unchanged. Please note that here the value of the threshold σ is just half of the corresponding value in $\chi^{2}$ Distribution Table. This is because the motion affinity is scaled by half in our implementation to reduce its dominance over the size affinity.Once a performance peak is identified at $\hat{\sigma }$, a fine grid search is performed on the set $\left\{\hat{\sigma }-0.2,\hat{\sigma }-0.1,\hat{\sigma },\hat{\sigma }+0.1,\hat{\sigma }+0.2\right\}$

The resulted value of σ on KITTI, NuScenes, and Waymo are respectively 6.5, 4.5, and 1.5.

### 4.2. Tracking Results

Evaluation Metrics: Classically, MOT systems are evaluated by the CLEAR MOT metrics [29] which compute tracking performance based on three cores quantities which are the number of False Positives, False Negatives, and ID Switches (the definition of these quantities can be found in Section 1). Intuitively, this set of metrics aims at evaluating a tracker’s precision in estimating tracks’ states as well as its consistency (i.e., keeping a unique ID for each even in the presence of occlusion). As pointed out by [30] and later by [6], there is a linear relation between MOTA and object detectors’ recall rate, as a result, MOTA does not provide a well-rounded evaluation performance of trackers. To remedy this, [6] proposes to average MOTA and MOTP over a range of recall rate, resulting in two integral metrics AMOTA and AMOTP which become the norm in recent benchmarks.

Datasets: To verify the effectiveness of our method, we benchmark it on three popular autonomous driving datasets which offer 3D MOT benchmark: KITTI, NuScenes, and Waymo. These datasets are collections of driving sequences collected in various environment using a multi-modal sensor suite including LiDAR. KITTI tracking benchmark interests in two classes of object which are cars and pedestrians. Initially, KITTI tracking was designed for MOT in 2D images and recently [6] adapts it to 3D MOT. NuScenes concerns a larger set of objects which comprises of cars, bicycles, buses, trucks, pedestrians, motorcycles, trailers. Waymo shares the same interest as NuScenes but groups car-like vehicles into a meta class.

Public Detection: As can be seen in Table 1, AMOTA highly depends on the precision of object detectors. Therefore, to have a fair comparison, the baseline detection results made publicly available by the benchmarks are used as the input to our tracking system. Specifically, we use Point-RCNN detection for KITTI dataset, MEGVII detection for NuScenes, and PointPillars with PPBA detection for Waymo.

The performance of our model compared to the SORT-style baseline model in three popular benchmarks are shown in Table 2.

As can be seen, our model consistently outperforms the baseline model in term of the primary metric AMOTA, thus proving the effectiveness of the 2-stage data association. Specifically, the improvements are 10.080%,3.922%, and 25.430% for KITTI, NuScenes and Waymo, respectively. It is worth noticing that our approach has more track fragmentations (FRAG), 259 compared to 93 of the base line, in KITTI. The reason for this is that at each time step tracklets have no matched detections are not reported by our approach, while the baseline predicts their pose using the constant velocity model (CV) and reports this prediction.

The comparison runtime on KITTI dataset of our tracking algorithm against AB3DMOT [6] is shown in Table 3. Despite the additional complexity added by the second stage of the data association (i.e., the Global Association step), our approach can achieve a runtime that is close to AB3DMOT on KITTI and exceeds the real-time speed by a large margin. On more challenging datasets, the object detector generates a significantly larger number of detections per frame on average, 57.50 on NuScenes and 264.18 on Waymo, compared to 10.04 of KITTI. This large number of detections enlarges the cost matrix of the Local and Global Association step, thus making the LAPs represented by them more costly to solve. Therefore, the runtime of our approach is reduced to 1.44 frames-per-second (fps) on NuScenes and 0.35 fps on Waymo. This runtime can be greatly improved if our approach is re-implemented in a compiling language such as C++.

The qualitative performance on NuScenes is illustrated by drawing the bird-eye view of a scene with tracking result, ground truth objects and detection result accumulated through time as in Figure 3 and Figure 4.

The difficulty of the 3D MOT can be appreciated by the noisy detection with several false positives denoted by the clutter in the top of Figure 4-Detection and false negatives, as shown by the absence of one trajectory in the top left corner of Figure 3-Detection.

### 4.3. Ablation Study

In this ablation study, the default method is the method presented in Section 3 which has

Two stages of data association (local and global). Each stage is formulated as a LAP and solved by a greedy matching algorithm [15].The affinity function the sum of position affinity and size affinity (as in Equation (Equation 4)).The motion model is Constant Turning Rate and Velocity (CTRV) for car-like objects (cars, buses, trucks, trailers, bicycles) and Constant Veloctiy (CV) for pedestrians.As mentioned in Section 4.1, the value of hyperparameters are set as follows: β=1.35 (in Equation (Equation 3)), tracklet confidence threshold $\tau^{c}=0.45$, and the affinity threshold σ=4.5 (in Equation (Equation 11))

To understand the effect of each component on the system’s general performance, we modify or remove each of them and carry out experiment with the rest of the system being kept the same as the default method and the same hyperparameters. The changes and the resulted performance are shown in Table 4.

It can be seen that solving the matching problem (formulated as a LAP) with the Hungarian algorithm instead of the greedy matching algorithm of [15] results in a marginal increase of AMOTA; however, this increased performance comes at the cost of increased execution time since the Hugarian algorithm has higher time complexity (cubic time compared to quadratic time.). In addition, using Constant Velocity model only reduces the AMOTA by 2.744% compared to the Default setting which shows the effectiveness of the Constant Turning Rate and Velocity model in predicting motion of car-like vehicle. Finally, performing global association only deteriorates the tracking performance confirms the importance of the local association step which significantly reduce the association ambiguity for the second stage.

## 5. Conclusions and Perspectives

In conclusion, this paper successfully adapted an image-based tracking method to the 3D space. Particularly, extensive experiments carried out in various datasets shows that our two-stage data association pipeline can result in significant improvement in the tracking accuracy by adding a certain degree of re-identification while keeping the added complexity to the minimum. Nevertheless, medium and long-term occlusion remains challenging for our approach due to the fact that the affinity function relies mostly on tracklets position whose prediction’s reliability reduces with the length of the prediction horizon. In the domain of image-based MOT, this problem is offend solved by exploiting tracklets’ appearance with Siamese networks [31,32]. However, the extension of this method to 3D space is not straightforward due to the lack of color and texture in point cloud. A possibility to resolve this issue is to associate 3D tracklets to 2D object detections, then carry out re-identification in images. Taking a different approach, a recent work in graph neural networks [33] proposes to jointly learn affinity function from point clouds and images.

## Figures and Tables

**Figure 1 sensors-21-02894-f001:**
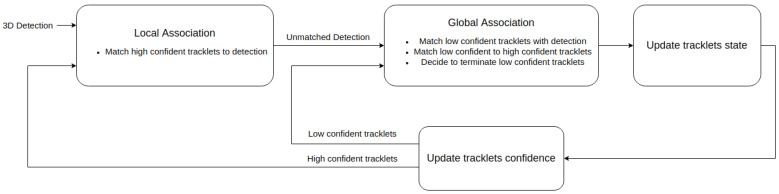
The pipeline of two-stage data association. The first stage—local association establish the correspondences between detections at this time step $D_{t}$ and high-confident tracklets $T_{0:t}^{h}$. Then, global association stage matches each low-confident tracklets $T^{i,l}$ with either a high-confident tracklet or a left-over detection, or terminates it.

**Figure 2 sensors-21-02894-f002:**
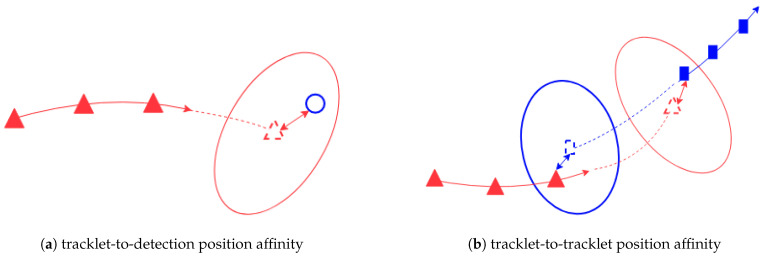
The computational scheme of position affinity. The filled triangles (or rectangles) are subsequent states of a tracklet. The colored arrow represents the time order: the closer to the tip, the more recent the state. The triangle (or rectangle) in dash line is the state propagated forward (or backward) in time. The covariance of these propagated states are denoted by ellipses with the same color. The two-headed arrows indicate the Mahalanobis distance. In the subfigure (**a**), the blue circle denotes a detection.

**Figure 3 sensors-21-02894-f003:**
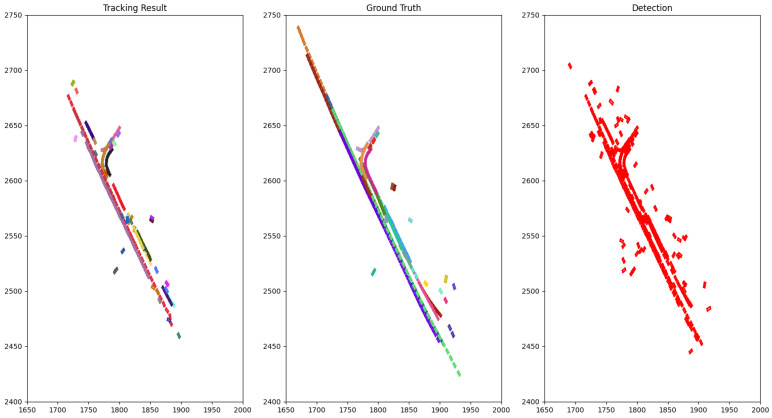
The bird-eye view of the tracking result for class car compared to the ground truth of scene 0796 (NuScenes) accumulated through time. Each rectangle represents a car and each color is associated with a track ID.

**Figure 4 sensors-21-02894-f004:**
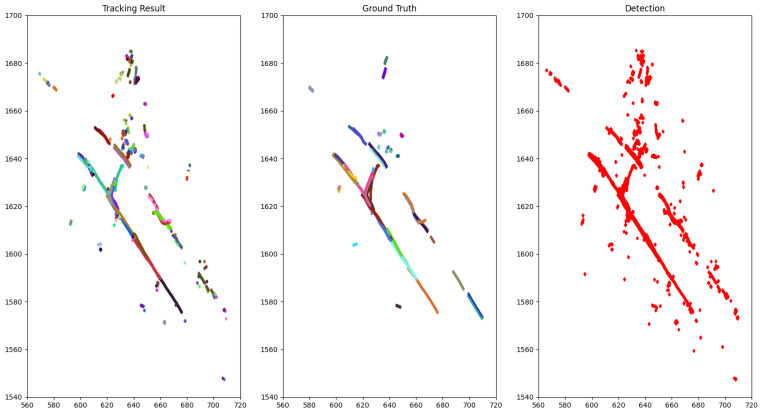
The bird-eye view of tracking result for class pedestrian compared to the ground truth of scene 0103 (NuScenes) accumulated through time. Each dot represents a pedestrian and each color is associated with a track ID.

**Table 1 sensors-21-02894-t001:** Summary of tracking methods which details are published in the leader board of NuScenes and Waymo Open Dataset.

Dataset	Method Name	Tracking Method	AMOTA	Object Detector	mAP
NuScenes	CenterPoint [9]	Greedy closest-point matching	0.650	CenterPoint	0.603
	PMBM	Poisson Multi-Bernoulli Mixture filter [14]	0.626	CenterPoint	0.603
	StanfordIPRL-TRI [15]	Hungarian algorithm with Mahalanobis distance as cost function and Kalman Filter	0.550	MEGVII [16]	0.519
	AB3DMOT [6]	Hungarian algorithm with 3D IoU as cost function and Kalman Filter	0.151	MEGVII	0.519
	CenterTrack	Greedy closest-point mathcing	0.108	CenterNet [17]	0.388
Waymo	HorizonMOT [18]	3-stage data associate, each stage is an assignment problem solved by Hungarian algorithm	0.6345	AFDet [19]	0.7711
	CenterPoint	Greedy closest-point matching	0.5867	CenterPoint	0.7193
	PV-RCNN-KF	Hungarian algorithm and Kalman Filter	0.5553	PV-RCNN [20]	0.7152
	PPBA AB3DMOT	Hungarian algorithm with 3D IoU as cost function and Kalman Filter	0.2914	PointPillars and PPBA [21]	0.3530

**Table 2 sensors-21-02894-t002:** Quantitative performance of our model on KITTI validation set, NuScenes validation set, and Waymo test set. AMOTA is the primary metric of these benchmarks. FP, FN IDS and FRAG are absolute numbers in the case of KITTI and NuScenes, while they are divided by the total number of objects in Waymo. The performance on Waymo is calculated at the difficulty of LEVEL 2.

Dataset	Method	AMOTA↑	AMOTP↓	MT↑	ML↓	FP↓	FN↓	IDS↓	FRAG↓
KITTI (val)	Ours	**0.415**	0.691	NA	NA	766	3721	10	259
	AB3DMOT [6]	0.377	**0.648**	NA	NA	**696**	**3713**	**1**	**93**
NuScenes (val)	Ours	**0.583**	**0.748**	**3617**	1885	13,439	**28,119**	**512**	**511**
	StanfordIPRL-TRI [15]	0.561	0.800	3432	**1857**	**12,140**	28,387	679	606
Waymo (test @ L2)	Ours	**0.365**	**0.263**	NA	NA	**0.089**	**0.533**	0.014	NA
	PPBA-AB3DMOT	0.291	0.270	NA	NA	0.171	0.535	**0.003**	NA

**Table 3 sensors-21-02894-t003:** Comparison of our approach’s runtime on KITTI dataset against AB3DMOT’s.

Class of Objects	Our Runtime (fps)	AB3DMOT’s Runtime (fps)
Car	115	186
Pedestrian	497	424
Cyclist	1111	1189

**Table 4 sensors-21-02894-t004:** Ablation study using NuScenes dataset.

Method	AMOTA↑	AMOTP↓	MT↑	ML↓	FP↓	FN↓	IDS↓	FRAG↓
Default	0.583	0.748	**3617**	1885	13,439	28,119	512	511
Hungarian for LAP	**0.587**	**0.743**	3609	**1880**	13,667	**28,070**	596	573
No ReID	0.583	0.748	3616	1882	13,429	28,100	**504**	**510**
Global assoc only	0.327	0.924	2575	2244	26,244	38,315	4215	3038
Const Velocity only	0.567	0.781	3483	1966	12,649	29,427	718	606
No size affinity	0.581	0.748	3595	1904	13,423	28,448	512	508
3D IoU as affinity	0.535	0.898	3090	2075	**9168**	33,041	550	528

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
