# Peer review of "A Two-Stage Data Association Approach for 3D Multi-Object Tracking"

_sensors, 2021, doi:10.3390/s21092894_

Round 1
Reviewer 1 Report
- In the definition of position affinity between a tracklet and a detection as shown in formula (5), what does the S stand for? And why use the last state of tracklet, instead of the whole tracklet? And there is another S in formula (7), which is stated as the covariance matrix, is it same with the one in formula (5)?
- To reduce the risk of false association, a threshold is applied to the affinity in formula (11), how to select this threshold and the effects on tracking results?
- For the ablation study, the reviewer wonder the performance when only local DA is used, other than global association only.
- By the way, how about the computing complexity for the proposed method?
The discussions on these issues will make the manuscript more convincing.
Reviewer 2 Report
The paper describes an approach for 3D Multi-Object tracking, originally developed by authors. The methodology could support the studies and industrial advances in autonomous car driving: the trajectory of other cars and pedestrian can be guessed from image analysis, also in case of occlusion or temporary loss of visibility. A two-stage data association is used to score the reliability of tracklets and update its state and confidence.
The paper is well structured and significant references have been included. The methodology description is supported by a strong mathematical background, including the dynamic model of both pedestrian and car/bus/truck motion.
The weak point of the manuscript is that the paper is very specialist and it cites methodologies which could be unknown to a reader that doesn’t work actively in the field. I suggest to include a few lines of comment of the main mathematical tools used:
In the text before Table 1: explain what is AMOTA.
Line 40: explain in a short sentence how AB3DMOT works.
Line 75 explain in a few lines what is the scope of the Hungarian algorithm and how it works.
Line 89: add a few lines to describe what is Mahalanobis distance
Line 97: could you describe Hankel matrix in a few words to provide the reader with an idea of this mathematical technique?
Line 118: could you define the function “argmax”? Is this a programming language standard function?
Line 229: could you summarize how CLEAR MOT works? Which kind of parameters does it keep into consideration? Just a few words to help the reader.
Some other remarks:
Check line 81 where probably you have to add a word before “the Hungarian algorithm”.
In lines 273-274 you define the value of some constants: are these values suggested by literature? Did you set it based on preliminary studies? How a change in these parameters affect the efficiency of the methodology?
What programming language did you set to carry out the implementation of your algorithm, validate it and produce Table 2 and 3?
Is your algorithm suitable for a real-time implementation? Which computational power should be expected to run your algorithm? Microcontroller (how many bytes of Flash memory and SRAM)? PLC? PC (which processor and RAM)?
Reviewer 3 Report
The paper presents a multi-object data assimilation algorithm focusing on road traffic tracking problems. The method uses a two-stage approach to assign measurements to tracks and track to tracks, based on confidence scores. The first and second section gives an adequate introduction and motivation and motivation to the subject. The results are clearly exposed and analyzed.
The English of the text is fine, some minor language errors need to be corrected in e.g. line 199 (picks, removes); line 173: (the a), line 168 (their).
Raised questions are the following:
- It would be nice to see the runtimes of the algorithms.
- What timestep was used?
- Line 276 states that the parameter beta is chosen empirically such that a high confident tracklet becomes low confident after being undetected for three frames. In this case could the tracklet-to-tracklet association (Fig.2b) consider more than one timestep? This may help reducing the track fragmentation.
Round 2
Reviewer 1 Report
All the concerned issues were cleared. No more comments.
Can be accepted after some writing language improving.